# Correlation between the Antibiotic Resistance Genes and Susceptibility to Antibiotics among the Carbapenem-Resistant Gram-Negative Pathogens

**DOI:** 10.3390/antibiotics10030255

**Published:** 2021-03-04

**Authors:** Salma M. Abdelaziz, Khaled M. Aboshanab, Ibrahim S. Yahia, Mahmoud A. Yassien, Nadia A. Hassouna

**Affiliations:** 1Department of Microbiology and Immunology, Faculty of Pharmacy, Ain Shams University, Organization of African Unity St., Abbassia, Cairo 11566, Egypt; salma_mustafa87@pharm.asu.edu.eg (S.M.A.); mahmoud.yassien@pharma.asu.edu.eg (M.A.Y.); nadia.hassouna@pharma.asu.edu.eg (N.A.H.); 2Research Center for Advanced Materials Science (RCAMS), King Khalid University, Abha 61413, Saudi Arabia; ihussien@kku.edu.sa; 3Advanced Functional Materials & Optoelectronic Laboratory (AFMOL), Department of Physics, Faculty of Science, King Khalid University, Abha 9004, Saudi Arabia; 4Nanoscience Laboratory for Environmental and Bio-Medical Applications (NLEBA), Semiconductor Lab., Physics Department, Faculty of Education, Ain Shams University, Roxy, Cairo 11757, Egypt

**Keywords:** carbapenem resistance, lower respiratory tract infections, *Klebsiella pneumoniae*, *Pseudomonas aeruginosa*, *Escherichia coli*, *ESBL*

## Abstract

In this study, the correlation between the antibiotic resistance genes and antibiotic susceptibility among the carbapenem-resistant Gram-negative pathogens (CRGNPs) recovered from patients diagnosed with acute pneumonia in Egypt was found. A total of 194 isolates including *Klebsiella pneumoniae* (89; 46%), *Escherichia coli* (47; 24%) and *Pseudomonas aeruginosa* (58; 30%) were recovered. Of these, 34 (18%) isolates were multiple drug resistant (MDR) and carbapenem resistant. For the *K. pneumoniae* MDR isolates (n = 22), *bla*_NDM_ (14; 64%) was the most prevalent carbapenemase, followed by *bla*_OXA-48_ (11; 50%) and *bla*_VIM_ (4; 18%). A significant association (*p* value < 0.05) was observed between the multidrug efflux pump (AcrA) and resistance to β-lactams and the aminoglycoside acetyl transferase gene (*aac-6’-Ib*) gene and resistance to ciprofloxacin, azithromycin and β-lactams (except for aztreonam). For *P. aeruginosa*, a significant association was noticed between the presence of the *bla*_SHV_ gene and the multidrug efflux pump (MexA) and resistance to fluoroquinolones, amikacin, tobramycin, co-trimoxazole and β-lactams and between the *aac-6’-Ib* gene and resistance to aminoglycosides. All *P. aeruginosa* isolates (100%) harbored the MexAB-OprM multidrug efflux pump while 86% of the *K. pneumoniae* isolates harbored the AcrAB-TolC pump. Our results are of great medical importance for the guidance of healthcare practitioners for effective antibiotic prescription.

## 1. Introduction

Gram-negative bacteria pose a significant treatment challenge to medical staff due to their widespread resistance to antibiotics. *Klebsiella pneumoniae* is a frequent human pathogen that causes many diseases, such as pneumonia, urinary tract infections and surgical wound infections, and serious life-threatening infections, such as endocarditis and septicemia. It can also cause necrotizing pneumonia and pyogenic liver abscesses [1]. It is responsible for about one-third of infections caused by all Gram-negative bacteria [2]. It is also frequently resistant to multiple antibiotics [3]. *Escherichia coli*, which is another member of the *Enterobacteriaceae* family, is the most common commensal in the gastrointestinal tract of people as well as an important pathogen. It can cause several diseases, including watery diarrhea, bloody diarrhea, urinary tract infections, acute neonatal meningitis and sepsis [4]. *Pseudomonas aeruginosa* is a Gram-negative bacterium that causes a myriad of diseases, especially in critically ill and immunocompromised patients. It is a common culprit of ventilator-associated pneumonia, urinary tract infections, skin and soft tissue infections and bacteremia [5].

There is no doubt that the discovery of antibiotics at the beginning of the 20th century has saved countless lives and revolutionized modern medicine. Unfortunately, the discovery of these “magic bullets” has been inevitably accompanied by the emergence of resistant pathogens [6]. Currently, medical experts are concerned about the return to the pre-antibiotic age [7]. From the analysis of the available bacterial genomes, it was found that more than 20,000 potential resistance genes already exist in a medical database [8].

The genetic origin of drug resistance differs among drug-resistant microorganisms; it can be due to either chromosomal or mobile genetic elements [9]. Resistance acquired from mobile genetic elements, such as plasmids and transposons, is more common than from those of the chromosome. A single plasmid can carry multiple genes which encode for resistance to several drugs, thus, it spreads multiple drug resistance among microorganisms [10]. Another major problem with plasmids is that they can cross many species and genus barriers, therefore, they allow resistance to spread in bacteria that are not necessarily exposed to antibiotics [11]. Thus, devastating consequences in human health emerge from the rapid and broad dissemination of resistance determinants by plasmids [12]. In general, bacteria use three main strategies to become resistant to different antibiotics: (a) preventing the drug from reaching its target (through reduced permeability or active efflux), (b) altering the drug target and (c) inactivating the antibiotic through antibiotic destruction or modification [10]. With the advance in molecular biology techniques, the resistance genes have been extensively studied and documented [13].

Efflux pumps, which are used by almost all bacterial cells to export toxic substances from the cell metabolism, can expel antibiotics from the cell as well [14]. Five families of bacterial drug efflux pumps have been previously identified [15]. In most cases, the efflux pumps are chromosomally encoded and therefore they are not easily transferable between bacteria [16]. In resistant Gram-negative bacteria, the widely spread multidrug efflux pumps are AcrAB-TolC and MexAB-OprM, which belong to the RND superfamily. These tripartite efflux transporters were first identified and characterized in *E. coli* and *P. aeruginosa*, respectively [17]. They are known to efflux antibiotics (β-lactams, fluoroquinolones, tetracycline and chloramphenicol), heavy metals, dyes, detergents and solvents, along with many other substrates [16]. 

The expression of hydrolytic enzymes known as β-lactamases is the most common mechanism of bacterial resistance to β-lactams. These enzymes specifically hydrolyze the β-lactam ring, leading to an inactivated product that cannot inhibit the bacterial transpeptidases any longer [10]. There are many β-lactamases encoded on mobile genetic elements, and this leads to their increased transmission and spread. Thus, it is very common to find bacteria harboring as many as eight different β-lactamases, and each one of them specifically inactivates a unique subset of β-lactam antibiotics [18]. It is important to have reliable and easily understandable nomenclature to refer to these enzymes, as more than 4300 unique enzymes have already been identified [19]. The β-lactamases are classified into four distinct classes based on their molecular structure: classes A through D. Classes A, C and D possess a serine residue at the active site to initiate bond hydrolysis, they are thus referred to as serine β-lactamases (SBLs). In contrast, the hydrolytic action of class B β-lactamases is facilitated by one or two essential zinc ions in the active sites and therefore they are known as metallo-β-lactamases (MBLs) [18]. 

Notable class A enzymes include (1) TEM, which is named for a patient called Temoniera and is the first plasmid-encoded β-lactamase identified in Gram-negative bacteria; (2) sulfhydryl variant (SHV) which is an enzyme with similar activity to TEM; (3) cefotaximase (CTX-M); and (4) *K. pneumoniae* carbapenemase (KPC), which is responsible for carbapenem resistance [19]. Class A extended-spectrum β-lactamases (ESBLs) of the TEM, SHV and CTX-M families are now among the most clinically significant β-lactamases that can hydrolyze penicillins and broad-spectrum cephalosporins, as well as monobactams [18]. The most clinically relevant and widespread members of class B enzymes are: (1) Verona integrin-encoded MBL (VIM); (2) imipenemase (IMP); and (3) New Delhi MBL (NDM), among others [20]. MBLs constitute the most molecularly diverse class of carbapenemases, and can hydrolyze nearly all β-lactams, except for monobactams. The class C enzymes (also referred to as AmpC enzymes) hydrolyze penicillins and cephalosporins. These enzymes are mainly chromosomally encoded; however, some have been reported to be carried on plasmids [18]. Class D SBLs include the oxacillinase (OXA) enzymes which are mainly plasmid encoded [18]. Recently, these enzymes have become increasingly important due to the ability of some members of this class to hydrolyze carbapenems along with other β-lactams, OXA-48 and related enzymes [21].

The enzymatic modification of the amino or hydroxyl groups of aminoglycosides is considered the major resistance mechanism to these antibiotics in Gram-negative and Gram-positive bacteria. This is facilitated by aminoglycoside phosphotransferases (APHs), aminoglycoside acetyltransferases (AACs) and/or aminoglycoside nucleotidyltransferases (ANTs) [10]. The structural modification of the aminoglycoside results in the inability of the modified antibiotic to bind to the target RNA due to unfavorable steric and/or electrostatic interactions [22]. Moreover, the modified variant of the enzyme, *aac*(*6*′)-*Ib*-*cr*, also exhibits a reduced quinolone susceptibility phenotype [23]. It has two amino acid changes, Trp102Arg and Asp179Tyr, which together allows the enzyme to acetylate and inactivate ciprofloxacin as well [24].

Therefore, through this research, we aim to develop an accurate local periodic report of antimicrobial resistance and to correlate the presence of certain antibiotic resistance gene(s) and susceptibility to antibiotics among the carbapenem-resistant Gram-negative pathogens (CRGNPs), particularly those conferring a multiple drug resistant (MDR) phenotype. The findings of this study will guide healthcare practitioners to more effective prescription patterns.

## 2. Results

The Gram-negative isolates collected in this study (n = 194) were *K. pneumoniae* (46%), *E. coli* (24%) and *P. aeruginosa* (30%). The antimicrobial susceptibility testing showed that the lowest resistance was observed to amikacin (15%), doxycycline (16%) and meropenem (18%). On the other hand, the highest resistance was observed to amoxicillin (79%), cefadroxil (78%), cefuroxime (78%) and cefotaxime (69%). It was found that 66.5% of the isolates (n = 129) were MDR. Appendix A shows the antibiogram results of the Gram-negative isolates in this study.

The meropenem-resistant isolates were selected for further study, as they are considered critical level priority pathogens according to the WHO [25]. None of the collected *E. coli* isolates were resistant to meropenem. On the other hand, 22 *K. pneumoniae* isolates (25%) and 12 *P. aeruginosa* isolates (21%) were resistant to meropenem. The results of the antibiogram results of the carbapenem-resistant isolates are shown in Appendix A. Detailed antibiogram results and genes detected in the carbapenem-resistant *K. pneumoniae* and *P. aeruginosa* isolates are shown in Appendix A, respectively. 

All the meropenem-resistant *K. pneumoniae* and *P. aeruginos* a isolates were resistant to cefotaxime and all of them contained one or more of the ESBL enzymes studied, however, only 16 *K. pneumoniae* (73%) and three *P. aeruginosa* (25%) isolates gave double disk synergy test (DDST). On the other hand, 19 meropenem-resistant *K. pneumoniae* (86%) and six meropenem-resistant *P. aeruginosa* (50%) harbored one or more of the carbapenemases studied, however, only 18 *K. pneumoniae* and three *P. aeruginosa* isolates gave positive modified Hodge test (MHT). Thus, the sensitivity of MHT for the *K. pneumoniae* isolates was almost 95% and for the *P. aeruginosa* isolates it was 50%. The results of the DDST and the MHT of the *K. pneumoniae* and *P. aeruginosa* isolates are shown in Appendix A, respectively. Several resistance genes were studied and their results are shown in Table 1, the number of resistance genes carried per resistant isolate is shown in Table 2 and Table 3 shows different antimicrobial resistance genotypes of the CRGNPs. 

As shown in Table 4, statistical analysis has shown a statistically significant association between the detection of resistance genes and the phenotypic antimicrobial resistance (*p* value ˂ 0.05). Calculation of the Pearson chi-square value showed a significant association between the presence of the *bla*_SHV_ gene and the multidrug efflux pump, MexA, in *P. aeruginosa* and resistance to fluoroquinolones, amikacin, tobramycin, co-trimoxazole and the β-lactams except for aztreonam. There was also a significant association between the presence of the *aac-6’-Ib* gene and resistance to aminoglycosides. Among the *K. pneumoniae* isolates, there was a significant association between the presence of the multidrug efflux pump, AcrA, and resistance to β-lactams except for cefepime and aztreonam. There was also a significant association between the presence of the *aac-6’-Ib* gene and resistance to ciprofloxacin, azithromycin and the β-lactams except for aztreonam. Statistical analysis has also shown a statistically significant association between the co-existence of antibiotic resistance genes on plasmids of the same isolate, including co-existence of *bla*_SHV_/*bla*_CTX-M_, *bla*_SHV_/*bla*_TEM_, *bla*_CTX-M_/*aac-6’-Ib* and *bla*_SHV_/*aac-6’-Ib*. Lastly, there was also a significant association between a positive MHT and resistance to meropenem. 

## 3. Discussion

Lower respiratory tract infections (LRTIs) are a global health concern as they are a leading cause of morbidity and mortality worldwide [26]. A study conducted in 2017 by the Global Burden of Diseases, Injuries, and Risk Factors reported that nearly 2.56 million deaths resulted from LRTIs, making them the fifth leading cause of death for all ages [27]. More than 50% of the deaths due to LRTIs were caused by bacteria for which antibiotics are commonly prescribed for treatment [28]. 

Antibiotic resistance is pernicious for both the health and economic wellbeing of societies; its threat to modern medicine has been internationally recognized and profusely addressed in recent years [29]. Resistant organisms cause infections that are more difficult to treat, as they require drugs that are often less easily available, more expensive and even more toxic [30]. Reports show that the likelihood of hospitalization and the duration of hospital stay were at least twice as great for patients infected with drug-resistant strains of the same organism [31]. 

Gram-negative pathogens are particularly disconcerting to medical staff as they are becoming increasingly resistant to all or nearly all available antibiotic options [32]. The emergence of MDR Gram-negative bacilli has affected almost every field of medicine [33]. The most severe Gram-negative infections are commonly caused by MDR *K. pneumoniae*, *E. coli*, *P. aeruginosa* and *Acinetobacter* [30]. The carbapenem-resistant Enterobacteriaceae, which are often referred to as “nightmare bacteria”, can survive and multiply in the sink drains of healthcare facilities. Thus, they inadvertently spread to patients and to the surrounding environment through wastewater [32]. These bacteria are medically alarming as they are resistant to carbapenems which are typically reserved as a last resort treatment option against drug-resistant pathogens [30].

*K. pneumoniae* was the most commonly isolated pathogen in our study. It was found that 77.5% of these isolates were MDR which was much higher than the results reported by Siwakoti et al. [34] of 28%. Fortunately, this was lower than the results reported by El-Sokkary et al. [35] of 89.6%. El-Sokkary et al. reported a similar resistance pattern to amikacin, cefuroxime, ciprofloxacin and levofloxacin. They reported much higher resistance to co-trimoxazole and co-amoxiclav. On the other hand, they reported lower resistance to cefotaxime, cefepime and meropenem. 

Comparing our results with an African systematic review [36], we found a similar prevalence of resistance to amoxicillin, co-amoxiclav, cefuroxime, amikacin, gentamicin, levofloxacin, tetracycline, co-trimoxazole and tobramycin. We observed higher resistance to meropenem, ciprofloxacin and cefotaxime. On the other hand, we observed much lower resistance to doxycycline. Comparing our results with a study carried out in China by Duan et al. [26], we found a similar prevalence of resistance to levofloxacin only. We observed higher resistance to cefepime, meropenem, amikacin, tobramycin, ciprofloxacin and co-trimoxazole. On the other hand, we observed much lower resistance to doxycycline. Comparing our results with the study reported by Singh et al. in India [37], we found a similar prevalence of resistance to meropenem, levofloxacin and co-trimoxazole. We observed much higher resistance to co-amoxiclav, cefotaxime, aztreonam and ciprofloxacin. On the other hand, we observed lower resistance to cefepime, amikacin and gentamicin. Comparing our results with another Indian study [38], we observed similar resistance to amikacin. We also observed lower resistance to co-amoxiclav, ciprofloxacin and gentamicin. However, we found much higher resistance to cefotaxime.

The antimicrobial susceptibility testing of the collected *E. coli* isolates showed that none of them was resistant to meropenem (0%). Approximately 60% of the isolates were MDR. Fortunately, this was lower than the results reported by another Egyptian study where 87.5% of the recovered *E. coli* isolates were MDR [35]. Comparing our results with the above study, we also observed a lower prevalence of resistance to amikacin, co-amoxiclav and co-trimoxazole. We found a similar resistance pattern to cefuroxime, cefepime and levofloxacin. Additionally, none of the *E. coli* isolates of the above study was resistant to meropenem. On the other hand, we observed higher resistance to cefotaxime and ciprofloxacin. 

Comparing our results with the previously mentioned systematic review of African countries [36], we found similar prevalence of resistance to amikacin, gentamicin, tobramycin, levofloxacin and meropenem. We observed higher resistance to cefuroxime, cefotaxime, cefepime and ciprofloxacin. On the other hand, we observed lower resistance amoxicillin, co-amoxiclav, co-trimoxazole, doxycycline and tetracycline. Comparing our results with the study reported in India by Singh et al. [37], we found similar prevalence of resistance to levofloxacin. We observed higher prevalence of resistance to co-amoxiclav, cefotaxime, aztreonam, ciprofloxacin and co-trimoxazole. On the other hand, we observed lower resistance to cefepime, meropenem, amikacin and gentamicin. Fortunately, while comparing our results with another Indian study [38], we observed much lower resistance to co-amoxiclav, amikacin, gentamicin and ciprofloxacin. However, we found higher resistance to cefotaxime and tetracycline.

*P. aeruginosa* usually shows resistance to multiple antibiotics, even those with considerable anti-pseudomonal activity. Therefore, it is better to treat infections caused by *P. aeruginosa* when guided by the susceptibility results of individual strains [35]. Approximately 55% of the isolates were MDR. Luckily, this was lower than the results reported in Egypt by El-Sokkary et al. [35], where 65.2% of the recovered *P. aeruginosa* isolates were MDR. Comparing our results with the above study, we also observed lower resistance to cefepime, meropenem, amikacin, ciprofloxacin and levofloxacin. 

Comparing our results with the aforementioned African systematic review [36], we found a similar prevalence of resistance to amikacin, gentamicin, amoxicillin, co-amoxiclav, cefotaxime, meropenem, ciprofloxacin and levofloxacin. We observed slightly lower resistance to co-trimoxazole and much lower resistance to tetracycline. Comparing our results with the previously mentioned study in China by Duan et al. [26], we observed a similar prevalence of resistance to cefepime, amikacin, gentamicin and levofloxacin. We observed higher resistance to tobramycin and ciprofloxacin. Fortunately, we observed lower resistance to meropenem. Comparing our results with the study reported in India by Singh et al. [37], we found a similar prevalence of resistance to aztreonam, amikacin and gentamicin. However, we observed much higher resistance to cefepime, meropenem, ciprofloxacin and levofloxacin. Fortunately, while comparing our results with another Indian study by Vijay and Dalela [38], we observed much lower resistance to all the comparable antimicrobials tested in the study: amikacin, gentamicin, co-amoxiclav, cefotaxime, ciprofloxacin and tetracycline. 

Although *K. pneumoniae* is usually resistant to amoxicillin, research on susceptibility to amoxicillin is still carried out in several countries around the world as not all *K. pneumoniae* isolates produce penicillinases [36]. This African systematic review included more than 144 studies and 149,000 samples from patients all across Africa. Likewise, several recent studies tested the susceptibility of *P. aeruginosa* to amoxicillin, co-amoxiclav and cefotaxime as not all *P. aeruginosa* isolates produce cephalosporinases [36,38,39].

All of our collected meropenem-resistant *K. pneumoniae* isolates were resistant to amoxicillin, co-amoxiclav, cefadroxil, cefuroxime and cefotaxime. Moreover, all of them (100%) were MDR. On the other hand, the lowest resistance was observed to doxycyline (4%; only one isolate was resistant). Comparing our results with a Chinese study on carbapenem-resistant *K. pneumoniae* [40], we observed similar resistance to cefotaxime, tobramycin, amikacin and co-trimoxazole. Fortunately, we found lower resistance to aztreonam, cefepime, ciprofloxacin, levofloxacin and gentamicin. Similarly, when comparing our isolates with another Chinese study [41], we observed similar resistance to cefuroxime, cefotaxime and levofloxacin, as well as to amikacin and gentamicin. However, we observed higher resistance to ciprofloxacin and lower resistance to co-trimoxazole. Comparing our study with a study covering 25 hospitals in China [42], we found a similar prevalence of resistance of the carbapenem-resistant *K. pneumoniae* isolates to cefotaxime, amikacin, tobramycin and co-trimoxazole. On the other hand, we observed lower resistance to cefepime, aztreonam, ciprofloxacin, levofloxacin and gentamicin.

All the tested meropenem-resistant *P. aeruginosa* isolates (100%) were resistant to amoxicillin, co-amoxiclav, cefadroxil, cefuroxime, cefotaxime, cefepime, ciprofloxacin and levofloxacin. Moreover, they were all MDR. Unfortunately, this is much higher than another study on carbapenem-resistant *Pseudomonas aeruginosa*; only 65% of the isolates were MDR [43]. The mechanisms of resistance of Gram-negative bacteria to carbapenems are complex; they are a result of the production of carbapenemases, a combination of porin loss or reduced expression with the production of ESBLs, or alteration of penicillin binding proteins (PBPs) at the drug action site of carbapenem [44]. Carbapenemase production is considered the most prevalent mechanism of resistance to carbapenems worldwide [45]. Carbapenem-resistant Enterobacteriaceae are designated by the CDC as nightmare bacteria, as carbapenem-resistant *K. pneumoniae* has a mortality rate of 40–50%, which is almost the same as that of Ebola virus, whose mortality rate is 50–60% [46].

There is a remarkable geographic variation in the distribution of different carbapenemases among carbapenem-resistant bacteria [46]. In some regions of the USA, KPCs comprise approximately 80% of the carbapenemases detected in *K. pneumoniae*, while MBLs are uncommonly detected in North America, with the exception of few imported cases [47]. KPCs also predominate in Italy, Portugal, Greece and China [48]. On the contrary, NDM is the most dominant carbapenemase in England [49], while OXA-48 is considered the most predominant carbapenemase detected in *K. pneumoniae* in Germany [49], Spain [50], Turkey [51] and Lebanon [52]. NDM, as well as OXA-48, predominate among the carbapenem-resistant bacteria in Saudi Arabia and the Gulf countries [53]. On the other hand, *K. pneumoniae* harboring both KPC and VIM together has been increasingly identified in Greece and France [49]. A study from Saudi Arabia and the Gulf countries [53] reported that the most frequently detected carbapenemases were OXA-48 (49%) and NDM (23%). None of the isolates produced KPC or VIM or IMP. However, Zhang et al. [42] from China reported that the most prevalent carbapenemase-producing gene in *K. pneumoniae* isolates was *bla*_KPC_ (77%), followed by *bla*_NDM_ (15%) and finally *bla*_IMP_ (2%). Another study [41] reported that the most prevalent gene was *bla*_OXA_ (42%), followed by *bla*_NDM_ (37%), followed by *bla*_KPC_ (17%) and finally *bla*_IMP_ (1%). A study conducted in India on carbapenem-resistant *P. aeruginosa* reported that the most frequently detected carbapenemase was *bla*_VIM_ (29%), followed *bla*_NDM_ (28%) and finally *bla*_SIM_ and *bla*_GIM_ (5% each) [54].

Patients suffering from community-acquired pneumonia usually receive empirical antimicrobial therapy, while the guidelines reserve microbiological testing for severe cases [55]. Standard microbiological identification techniques, followed by antimicrobial susceptibility testing, and then followed by PCR identification of the resistance genes of concern is a tedious process that requires several days. This delay exposes the patients to the unnecessary adverse effects of the drugs, as well as extending the hospital stay for complicated cases, which increases the risk that the patients contract a hospital-acquired infection [56]. It is extremely important to implement rapid techniques that allow the identification of the causative pathogens within a few hours. This would ensure more effective antimicrobial therapy within a few hours following the diagnosis [55]. One of these techniques is the Biofire^®^ FilmArray^®^ Pneumonia Panel which accurately identifies 33 targets in sputum and bronchoalveolar lavage samples in about one hour. It is a multiplex PCR technology that contains probes for eight respiratory viruses, 18 bacteria and seven clinically relevant resistance genes (*mecA*/*C*, *bla*_KPC_, *bla*_NDM_, *bla*_VIM_, *bla*_OXA-48-like_, *bla*_IMP_ and *bla*_CTX-M_). This technology identifies the nucleic acids in the samples even if the pathogen is fastidious or the patient received prior antimicrobial therapy which would render the culture results incomprehensive [57]. Other rapid molecular diagnosis techniques include the RespiFinder^®^ SMART 22 FAST, the Unyvero pneumonia cartridge, the ResPlex^TM^ Panels, scalable target analysis routine (STAR) technology and PLEX-ID technology [58,59,60,61]. Unfortunately, these techniques are not widespread in Egyptian hospitals as they are much more expensive.

In conclusion, the results obtained in this study are of great and relevant medical importance to healthcare practitioners for effective and proper antibiotic prescription. Investment to incorporate the rapid identification techniques in Egyptian hospitals should become a medical priority to allow improved routine care.

## 4. Materials and Methods

### 4.1. Microorganisms

A total of 194 clinical Gram-negative bacterial isolates were recovered from sputum clinical specimens discharged from the microbiology laboratory at Al-Demerdash Hospital, Cairo, Egypt from patients suffering from acute lobar pneumonia according to hospital records during the period from January 2018 to February 2019. Only patients who did not receive previous antimicrobial treatment were included in the study. The isolates were identified using conventional microbiological techniques. Further confirmation of some of the results was done using the API^®^ 20E identification kit (bioMérieux, Lyon, France). *Escherichia coli* ATCC^®^ 25922, *E. coli* ATCC^®^ 35218, *K. pneumoniae* ATCC^®^ 700603 were used in the quality control of antimicrobial disk diffusion susceptibility tests. The whole study was approved by the Faculty of Pharmacy, Ain Shams University Research Ethics Committee (ENREC-ASU-Nr. 94) where both informed and written consent was obtained from patients or parents of patients after explaining the study purpose.

### 4.2. Antimicrobial Susceptibility Test

The antimicrobial susceptibilities, including the Kirby–Bauer disk diffusion method and minimum inhibitory concentration (MIC), were tested as recommended by the Clinical and Laboratory Standard Institute (CLSI) [62]. Disks were obtained from Oxoid^®^, UK and Bioanalyse^®^, Turkey. The tested antimicrobials were: amikacin (30 µg), gentamicin (10 µg), tobramycin (10 µg), amoxicillin (25 µg), amoxicillin/clavulanic acid (20/10 µg), cefadroxil (30 µg), cefuroxime (30 µg), cefotaxime (30 µg), cefepime (30 µg), meropenem (10 µg), aztreonam (30 µg), ciprofloxacin (5 µg), levofloxacin (5 µg), doxycycline (30 µg), tetracycline (30 µg), trimethoprim/sulfamethoxazole SXT (1.25/23.75 µg) and azithromycin (15 µg). The MIC microdilution broth test was done in triplicate (CLSI) [62]. Isolates resistant to at least one antimicrobial agent in 3 or more antimicrobial categories are considered MDR. This is standardized international terminology proposed by the Centers for Disease Control and Prevention (CDC) and the European CDC [63].

### 4.3. Phenotypic Detection of ESBLs

We performed the double disk synergy test (DDST) developed by Jarlier and co-workers to detect potential ESBL producers [64]. A fresh inoculum of the potential isolate was prepared in isotonic saline to match the turbidity of 0.5 McFarland standard suspension. Then, the surface of a Mueller–Hinton agar plate was swabbed in three different directions and along the rim of the plate. Disks containing 30 μg of cefotaxime, cefepime and aztreonam were placed 20 mm apart, center to center, from a disk containing amoxicillin/clavulanic acid (20/10 μg) on the surface of the inoculated plate and incubated at 37 °C for 16 to 18 hours. The presence of ESBL was indicated by the enhancement of the inhibition zones between any of those disks towards the disk containing clavulanic acid.

### 4.4. Phenotypic Detection of Carbapenemases

We performed the modified Hodge test (MHT) to detect the presence of carbapenemase. Molecular detection of the carbapenemase-coding genes remains the most specific method of detection, however, it is the most expensive option and it is susceptible to false negatives if the specific carbapenemase gene present in the isolate is not targeted. The test was done according to the CLSI guidelines [62]. The plate was examined for enhanced growth of *E. coli* ATCC^®^ 25,922 around the test isolate streak at the intersection of the streak and the inhibition zone.

### 4.5. Detection of Selected Resistance Genes

The CRGNPs were selected for further study as they are categorized as critical priority pathogens. This was done according to the antibiotic-resistant priority pathogens that pose the greatest threat to human health, published by the WHO [25]. The genomic DNA was extracted using a Genomic DNA Extraction Kit (Thermo Scientific, Waltham, MA, USA) and the plasmid DNA was extracted using a GeneJet Plasmid Miniprep Kit (Thermo Scientific, USA). The extracted DNA was used as the template in the polymerase chain reaction (PCR) amplification cycles. The PCR products were detected by agarose gel electrophoresis [65]. The primers (oligonucleotides) used to amplify the studied resistance genes are listed in Table 5. Some PCR products were purified and sequenced at GATC Biotech Company (Constance, Germany) through Sigma Scientific Services Company (Cairo, Egypt) by the use of an ABI 3730xl DNA Sequencer. The products were analyzed and assembled using the Staden Package program version 3 (http://staden.sourceforge.net/ (accessed on 20 November 2020)). Finally, they were submitted in the NCBI GenBank database and their corresponding accession codes were obtained. The accession codes of the genes detected in this study are shown in Appendix A.

### 4.6. Statistical Analysis

Statistical analysis of the data was performed using IBM SPSS Statistics software for Windows v.20.0 (IBM Corp., Armonk, NY, USA). Qualitative data were expressed as frequency and percentage. A chi-square test was used to compare categorical variables. All tests were two-tailed, and *p*-value < 0.05 was considered as statistically significant.

## 5. Conclusions

The results of our study highlight the extensive spread of resistant pathogens in our community. This calls for strenuous regulations to rationalize antibiotic prescription and eliminate over-the-counter antibiotic dispensing. Improved diagnostic tests to determine the etiology of LRTIs would allow more judicious use of antibiotics. This would decrease the risk of propagating antimicrobial resistance, as well as the unwanted adverse effects of antibiotics, including the development of *Clostridium difficile*.

## Figures and Tables

**Table 1 antibiotics-10-00255-t001:** Resistance genes detected in carbapenem-resistant Gram-negative pathogens (CRGNPs).

Gene	*K. pneumoniae* (n = 22) n° (%)	*P. aeruginosa* (n = 12) n° (%)
*bla* _KPC_	0 (0)	0 (0)
*bla* _IMP_	0 (0)	0 (0)
*bla* _VIM_	4 (18)	3 (25)
*bla* _NDM_	14 (64)	1 (8)
*bla* _OXA_	11 (50)	3 (25)
*bla* _CTX-M_	15 (68)	8 (67)
*bla* _SHV_	10 (45)	11 (92)
*bla* _TEM_	10 (45)	7 (58)
*aac(6’)-Ib-cr*	20 (91)	10 (83)
*mexA*	-	12 (100)
*acrA*	19 (86)	-

n°: number of isolates carrying the gene, %: approximate percentage.

**Table 2 antibiotics-10-00255-t002:** Number of resistance genes carried per resistant isolate.

n° of Resistance Genes/Isolate	*K. pneumoniae*	*P. aeruginosa*	Total Isolates	
			n°	%
7			3	9
6	6	2	8	23
5	4	6	10	29
4	3	2	5	15
3	3	1	4	12
2	2	1	3	9
1	1	—	1	3

n°: number of isolates carrying the genes, %: approximate percentage.

**Table 3 antibiotics-10-00255-t003:** Antimicrobial resistance genotypes of CRGNPs (n = 34)**.**

Genotype	No.	≈%
*mex*A/*bla*_CTX-M_/*bla*_SHV_/*bla*_TEM_/*aac(6’)-Ib*	5	14
*acr*A/*bla*_OXA_/*bla*_CTX-M_/*bla*_SHV_/*bla*_TEM_/*aac(6’)-Ib* or *mex*A/*bla*_OXA_/*bla*_CTX-M_/*bla*_SHV_/*bla*_TEM_/*aac(6’)-Ib*	3	8
*acr*A/*bla*_NDM_/*bla*_OXA_/*bla*_CTX-M_/*bla*_SHV_/*aac(6’)-Ib* or *mex*A/*bla*_NDM_/*bla*_OXA_/*bla*_CTX-M_/*bla*_SHV_/*aac(6’)-Ib*	2	6
*acr*A/*bla*_NDM_/*bla*_CTX-M_/*bla*_SHV_/*aac(6’)-Ib*	2	6
*acr*A/*bla*_NDM_/*bla*_OXA_/*bla*_CTX-M_/*bla*_SHV_/*bla*_TEM_/*aac(6’)-Ib*	1	3
*acr*A/*bla*_VIM_/*bla*_NDM_/*bla*_CTX-M_/*bla*_SHV_/*bla*_TEM_/*aac(6’)-Ib*	1	3
*acr*A/*bla*_VIM_/*bla*_OXA_/*bla*_CTX-M_/*bla*_SHV_/*bla*_TEM_/*aac(6’)-Ib*	1	3
*acr*A/*bla*_VIM_/*bla*_NDM_/*bla*_SHV_/*bla*_TEM_/*aac(6’)-Ib*	1	3
*acr*A/*bla*_NDM_/*bla*_CTX-M_/*bla*_SHV_/*bla*_TEM_/*aac(6’)-Ib*	1	3
*acr*A/*bla*_NDM_/*bla*_OXA_/*bla*_CTX-M_/*bla*_TEM_/*aac(6’)-Ib*	1	3
*acr*A/*bla*_NDM_/*bla*_OXA_/*bla*_CTX-M_/*aac(6’)-Ib*	1	3
*acr*A/*bla*_OXA_/*bla*_CTX-M_/*bla*_TEM_/*aac(6’)-Ib*	1	3
*mex*A/*bla*_OXA_/*bla*_SHV_/*bla*_TEM_/*aac(6’)-Ib*	1	3
*acr*A/*bla*_VIM_/*bla*_NDM_/*aac(6’)-Ib*	1	3
*bla*_NDM_/*bla*_OXA_/*bla*_CTX-M_/*aac(6’)-Ib*	1	3
*acr*A/*bla*_NDM_/*bla*_OXA_/*aac(6’)-Ib*	1	3
*mex*A/*bla*_VIM_/*bla*_SHV_/*aac(6’)-Ib*	1	3
*mex*A/*bla*_CTX-M_/*bla*_SHV_/*aac(6’)-Ib*	1	3
*acr*A/*bla*_CTX-M_/*aac(6’)-Ib*	1	3
*acr*A/*bla*_NDM_/*bla*_CTX-M_	1	3
*bla*_NDM_/*bla*_CTX-M_/*bla*_TEM_	1	3
*mex*A/*bla*_VIM_/*bla*_SHV_	1	3
*bla*_OXA_/*aac(6’)-Ib*	1	3
*acr*A/*aac(6’)-Ib*	1	3
*mex*A/*bla*_VIM_	1	3
*acr*A	1	3

**Table 4 antibiotics-10-00255-t004:** Statistical association between genotype and minimum inhibitory concentration (MIC) of the antibiotics and their respective *p* values.

Significant Associations (Genotype and MIC of the Antibiotic)	Pearson Chi-Square
*bla*_SHV_/amoxicillin	0.015
*bla*_SHV_/co-amoxiclav	0.015
*bla*_SHV_/cefadroxil	0.015
*bla*_SHV_/cefuroxime	0.015
*bla*_SHV_/cefotaxime	0.015
*bla*_SHV_/cefepime	0.015
*bla*_SHV_/meropenem	0.015
*bla*_SHV_/ciprofloxacin	0.015
*bla*_SHV_/levofloxacin	0.015
*bla*_SHV_/amikacin	0.019
*bla*_SHV_/tobramycin	0.00
*bla*_SHV_/co-trimoxazole	0.049
*mex*A/amoxicillin	0.00
*mex*A/co-amoxiclav	0.00
*mex*A/cefadroxil	0.00
*mex*A/cefuroxime	0.00
*mex*A/cefotaxime	0.00
*mex*A/cefepime	0.00
*mex*A/meropenem	0.00
*mex*A/ciprofloxacin	0.00
*mex*A/levofloxacin	0.00
*mex*A/tobramycin	0.015
*mex*A/co-trimoxazole	0.002
*aac6’-Ib*/amikacin	0.040
*aac6’-Ib*/gentamicin	0.012
*aac6’-Ib*/tobramycin	0.005
*bla*_SHV_/amikacin	0.019
*bla*_SHV_/tobramycin	0.000
*bla*_CTX-M_/gentamicin	0.015
*bla*_TEM_/gentamicin	0.002
*acr*A/amoxicillin	0.026
*acr*A/co-amoxiclav	0.026
*acr*A/cefadroxil	0.026
*acr*A/cefuroxime	0.026
*acr*A/cefotaxime	0.026
*acr*A/meropenem	0.026
*aac6’-Ib*/ciprofloxacin	0.012
*aac6’-Ib*/azithromycin	0.005
*aac6’-Ib*/amoxicillin	0.008
*aac6’-Ib*/co-amoxiclav	0.008
*aac6’-Ib*/cefadroxil	0.008
*aac6’-Ib*/cefuroxime	0.008
*aac6’-Ib*/cefotaxime	0.008
*aac6’-Ib*/meropenem	0.008
*bla*_SHV_/*bla*_CTX-M_	0.047
*bla*_SHV_/*bla*_TEM_	0.036
*bla*_CTX-M_/*aac6’-Ib*	0.027
*bla*_SHV_/*aac6’-Ib*	0.016
Modified Hodge test/meropenem	0.001

MIC: minimum inhibitory concentration.

**Table 5 antibiotics-10-00255-t005:** Primers used in this study, the target resistance genes, the expected product sizes (bp), the used annealing temperatures (T_a_) and their references.

Target Gene	Primer Sequence (5’→3’)		Size (bp)	T_a_ (°C)	Reference
*bla* _KPC_	P_f_	TGTCACTGTATCGCCGTC	1100	50	[66]
	P_r_	CTCAGTGCTCTACAGAAAACC			
*bla* _IMP_	P_f_	CTACCGCAGCAGAGTCTTTG	587	50	[67]
	P_r_	AACCAGTTTTGCCTTACCAT			
*bla* _VIM_	P_f_	TCTACATGACCGCGTCTGTC	748	50	[68]
	P_r_	TGTGCTTTGACAACGTTCGC			
*bla* _NDM_	P_f_	GGTTTGGCGATCTGGTTTTC	621	50	[69]
	P_r_	CGGAATGGCTCATCACGAT			
*bla* _OXA_	P_f_	GCGTGGTTAAGGATGAACAC	438	50	[70]
	P_r_	CATCAAGTTCAACCCAACCG			
*bla* _CTX-M_	P_f_	CGCTTTGCGATGTGCAG	550	50	[71]
	P_r_	ACCGCGATATCGTTGGT			
*bla* _SHV_	P_f_	GGTTATGCGTTATATTCGCC	867	50	[72]
	P_r_	TTAGCGTTGCCAGTGCTC			
*bla* _TEM_	P_f_	ATGAGTATTCAACATTTCCG	867	50	[72]
	P_r_	CTGACAGTTACCAATGCTTA			
*aac(6’)-Ib-cr*	P_f_	TTGCGATGCTCTATGAGTGG	358	50	[73]
	P_r_	CGTTTGGATCTTGGTGACCT			
*mexA*	P_f_	CGACCAGGCCGTGAGCAAGCAGC	316	65	[74]
	P_r_	GGAGACCTTCGCCGCGTTGTCGC			
*acrA*	P_f_	ATCAGCGGCCGGATTGGTAAA	312	50	[75]
	P_r_	CGGGTTCGGGAAAATAGCGCG			

## Data Availability

Data are available within the article and all gene sequences are available with the NCBI accession codes provided within this manuscript.

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
