# Peer review of "Correlation between the Antibiotic Resistance Genes and Susceptibility to Antibiotics among the Carbapenem-Resistant Gram-Negative Pathogens"

_antibiotics, 2021, doi:10.3390/antibiotics10030255_

Round 1

Reviewer 1 Report

This manuscript by Abdelaziz et al highlights the correlation between antibiotic susceptibility and antibiotic resistance genes among carbapenem-resistant Gram-negative pathogens. The authors used multidrug-resistant and carbapenem resistance of E. coli, P. aeruginosa and K. pneumonia from 194 clinical Gram-negative bacterial isolates.

Overall, this is a well-written and straightforward article with a clear objective. I have a few comments related to the presentation of the results:

Specific comments:

  1. There are too many tables in the main text. Some of the tables can be supplementary. The authors could consider moving  Table 1-3, Table 7-9 to the supplementary data.
  2. The authors should report MICs for E. coli isolates as well.
  3. Table 8, the representation of the first column "significant associations" is misleading. This column shows the association between genotype and MIC of the antibiotic [not the antibiotic]. The column name should be changed accordingly. Perhaps authors could consider presenting this information in a plot rather than a table.
  4. Line 76; italics  E. coli and P. aeruginosa.

Author Response

Reviewer 1 comments:

This manuscript by Abdelaziz et al highlights the correlation between antibiotic susceptibility and antibiotic resistance genes among carbapenem-resistant Gram-negative pathogens. The authors used multidrug-resistant and carbapenem resistance of E. coliP. aeruginosa and K. pneumoniae from 194 clinical Gram-negative bacterial isolates. Overall, this is a well-written and straightforward article with a clear objective. I have a few comments related to the presentation of the results:

  1. There are too many tables in the main text. Some of the tables can be supplementary. The authors could consider moving Table 1-3, Table 7-9 to the supplementary data.

Authors’ response:

  1. The authors should report MICs for  coli isolates as well.

Authors’ response:

The MIC reported for the K. pneumoniae and P. aeruginosa was for meropenem. None of the E. coli isolates were resistant to meropenem by disk diffusion method and all of them had MIC ≤ 1µg/ml, and therefore these isolates were excluded from further study on carbapenem resistance..

  1. Table 8, the representation of the first column "significant associations" is misleading. This column shows the association between genotype and MIC of the antibiotic . The column name should be changed accordingly. Perhaps authors could consider presenting this information in a plot rather than a table.

Authors’ response:

This comment is sincerely appreciated. Table 8 has been renamed table 4 and we changed the title of the first column "significant associations" into association between genotype and MIC of the antibiotic according to reviewer’s recommendation.

  1. Line 76; italics  E. coli and P. aeruginosa.

Authors’ response:

Corrected in the revised manuscript. Page 2, line 76. 

Reviewer 2 Report

The authors conducted in a single hospital a descriptive study of the antimicrobial susceptibility and presence of resistance genes in some Gram-negative bacteria (i.e. E. coli, K. pneumoniae, and P. aeruginosa) isolated from patients with LRTIs. For all bacterial isolates included, they searched a selection of antimicrobial resistance genes including carbapenemase genes.

Methods

The authors justified that they used the modified Hodge test for detecting carbapenemases by the fact that molecular methods for screening carbapenemases are expensive and do not allow to identify all carbapenemase genes. However, apart molecular methods, others are currently commonly used like immunochromatography or combined disks, which are probably easier to interpret.

It seems that the authors did not use antimicrobial resistance criteria for undertaking resistance gene detection. Maybe it would have been more rational to select the gene detection according to the antimicrobial resistance patterns of the isolates.  

Additional data about the LRTIs should be provided. For example, what was the proportion of ventilator-associated pneumonia?  This is important to understand why the authors did not include A. baumannii in their study.

Results

I do not understand certain results in table 1. The authors identified 15% of K.pneumoniae which were susceptible to amoxicillin.  However, the se bacteria are known to be intrinsically resistant to amoxicillin by producing a penicillinase. Similarly, 16% of P. aeruginosa isolates were susceptible to amoxicillin, 36% were susceptible to co-amoxiclav, and 55% were susceptible to cefotaxime. However, P. aeruginosa is intrinsically resistant to those antibiotics by producing a cephalosporinase…

Line 173: the low percentage of positive DDST in P. aeruginosa isolates producing an ESBL is not surprising. Indeed, the synergy can be hidden by the overproduction of cephalosporinase.

Line 189: Is it really surprising that there was a statistically significant association between the detection of resistance genes and the phenotypic antimicrobial resistance?

Line 202: Similarly, even though the MHT is maybe not the most relevant test to identify carbapenemase production, it seems logical that there was a significant association between a positive MHT test and resistance to meropenem.

Line 235: “77.5% of K. pneumoniae isolates were MDR”. In the methods section, the MDR criteria is not defined.

Overall, the study reported local data, which confirm that antimicrobial resistance is widespread. However, this data is well-known as numerous papers concerning this public health concern have already been published. In addition, the data of antimicrobial susceptibility would be enough to alert the clinicians about this problem. Identifying resistance genes had mainly a descriptive interest.

The conclusions are not completely supported by the data. Although there is a high proportion of resistant strains, it seems difficult to claim that infection control practices are lacking without study data or references about the compliance with these practices at a local or a national level.

Author Response

Reviewer 2 comments:

The authors conducted in a single hospital a descriptive study of the antimicrobial susceptibility and presence of resistance genes in some Gram-negative bacteria (i.e. E. coliK. pneumoniae, and P. aeruginosa) isolated from patients with LRTIs. For all bacterial isolates included, they searched a selection of antimicrobial resistance genes including carbapenemase genes.

Authors’ response:

A single hospital was included in the study; as Al-Demerdash Hospital, Cairo, Egypt is one of the two biggest hospitals in the country; it is the educational hospital of Ain Shams university, the university is ranked first in Egypt according to the most recent Academic Ranking of World Universities (Shanghai Ranking). Patients from all over Egypt come to seek medical consultation and treatment as the hospital includes some of the highest esteemed medical staff. Thus, although it is a single hospital study, it is highly reputable and covers a large demographic of patients. Every year, more than 1 million and a half patients visit the clinics of the hospital from which 220,000 patients are admitted. More than 160,000 medical operations are carried out yearly. Therefore, data reported from Al-Demerdash is of big medical significance in our country,

Methods

The authors justified that they used the modified Hodge test for detecting carbapenemases by the fact that molecular methods for screening carbapenemases are expensive and do not allow to identify all carbapenemase genes. However, apart molecular methods, others are currently commonly used like immunochromatography or combined disks, which are probably easier to interpret.

Authors’ response:

We thank the reviewer for his comment. However, the modified Hodge test has been carried out as well as other most common phenotypic testes including, the antimicrobial susceptibility to different antimicrobials including different carbapenems. In addition, molecular detection via PCR followed by DNA sequencing have been performed in this study and the obtained consensus sequences of the different carbapenemases have been submitted into the GenBank database (table 5S, supplementary). However, the tests pointed out be the reviewers will be considered in future study.

It seems that the authors did not use antimicrobial resistance criteria for undertaking resistance gene detection. Maybe it would have been more rational to select the gene detection according to the antimicrobial resistance patterns of the isolates.

Authors’ response:

The resistance gene detection was accomplished based on the antibiogram analysis conducted in this study for different antimicrobial agents. For example, ESBLs (blaSHV, blaCTX-M, blaTEM) detection was carried out on the isolates that exhibited resistance to beta lactams including, 3rd and 4th generation cephalosporins. Also aac(6’)-Ib-cr  was detected in isolates that showed resistance to fluoroquinolones and or aminoglycosides. The isolates selected for further study were the meropenem-resistant isolates as these isolates were the most medically alarming isolates according to the WHO report of 2017. We investigated these isolates for several frequently encountered resistance genes and tried to make a correlation between the presence of these genes and carbapenem resistance which is considered a medical emergency as carbapenems are considered last treatment option against MDR bacteria. The findings of this study are of great medical importance for guidance the healthcare practitioners for effective antibiotic prescription

Additional data about the LRTIs should be provided. For example, what was the proportion of ventilator-associated pneumonia?  This is important to understand why the authors did not include A. baumannii in their study.

Authors’ response:

All isolates were recovered from patients suffering from acute community-acquired lobar pneumonia; either out-patients or those requiring medical admission. However, none of the isolates were ventilator-associated, as those isolates would require a different perspective of research, along with different suggested antibiotics for susceptibility testing. This information was clarified and highlighted in the revised manuscript; (page 10, Lines 322-326). We also focused on patients who did not receive previous antimicrobial treatment as mentioned in the materials and methods section. None of the recovered isolates was A. baumannii, and this is not uncommon in Egypt where a recent study on community-acquired pneumonia also lacked any A. baumannii isolates [1].

Results

I do not understand certain results in table 1. The authors identified 15% of K.pneumoniae which were susceptible to amoxicillin.  However, these bacteria are known to be intrinsically resistant to amoxicillin by producing a penicillinase. Similarly, 16% of P. aeruginosa isolates were susceptible to amoxicillin, 36% were susceptible to co-amoxiclav, and 55% were susceptible to cefotaxime. However, P. aeruginosa is intrinsically resistant to those antibiotics by producing a cephalosporinase.

Authors’ response:

We thank the reviewer for this comment. Actually, a recent African systematic review tested the susceptibility of K. pneumoniae to amoxicillin as not all K. pneumoniae isolates produce penicillinases [2]. This review included more than 144 studies and 149,000 samples from patients all across Africa. However, all the meropenem-resistant K. pneumoniae isolates in our study were resistant to amoxicillin. Several recent studies tested the susceptibility of P. aeruginosa to amoxicillin, co-amoxiclav and cefotaxime as not all P. aeruginosa isolates produce cephalosporinases [references 2–4]. However, all the meropenem-resistant P. aeruginosa isolates in our study were resistant to amoxicillin, co-amoxiclav and cefotaxime.

Line 173: the low percentage of positive DDST in P. aeruginosa isolates producing an ESBL is not surprising. Indeed, the synergy can be hidden by the overproduction of cephalosporinase.

Authors’ response:

This is indeed true. Reported positive DDST in P. aeruginosa isolates rarely exceeds 33% even when the disks are placed at 20-mm distance from clavulanic acid [5]. However, the DDST is the most widely used phenotypic test to detect ESBLs in Egypt, due to the ease of its application and relatively cheap materials required. Therefore, it must be noted that when it is the case of P. aeruginosa isolates, it is not an accurate prediction of presence of ESBLs.

Line 189: Is it really surprising that there was a statistically significant association between the detection of resistance genes and the phenotypic antimicrobial resistance?

Authors’ response:

The results of the statistical analysis are only listed for definitive interpretation of the results. The significant associations found through the statistical analysis only confirm that the choice of the genes studied had clear relation with the resistance to these antibiotics. Thus, bacteria found resistant to these antibiotics are most likely to carry these resistance genes even without carrying out PCR and sequencing.

Line 202: Similarly, even though the MHT is maybe not the most relevant test to identify carbapenemase production, it seems logical that there was a significant association between a positive MHT test and resistance to meropenem.

Authors’ response:

It is indeed true and it is the reason that we listed the association between positive MHT and resistance to meropenem. Funding of national hospitals in Egypt, or many developing countries, cannot afford detecting carbapenemase in all the suspected isolates by PCR. The MHT is much less expensive for application and yet it is highly sensitive. Thus with a significant association, the microbiologist in the hospital laboratory can have confidence in the result obtained and hastily report to the medical staff to switch to the most appropriate treatment options still available.

Line 235: “77.5% of K. pneumoniae isolates were MDR”. In the methods section, the MDR criteria is not defined.

Authors’ response:

Isolates resistant to at least one antimicrobial agent in 3 or more antimicrobial categories are considered MDR. This is standardized international terminology proposed by the Centers for Disease Control and Prevention (CDC) and the European CDC [6]. Corrected in the resubmitted manuscript. (Page 11, Lines 344-347).

Overall, the study reported local data, which confirm that antimicrobial resistance is widespread. However, this data is well-known as numerous papers concerning this public health concern have already been published. In addition, the data of antimicrobial susceptibility would be enough to alert the clinicians about this problem. Identifying resistance genes had mainly a descriptive interest.

Authors’ response:

We thank the reviewer’sfor his valuable comment. This is indeed true that the data of antimicrobial resistance is enough nowadays to alert the clinicians and guide to most appropriate treatment options. However, genes detected had mainly research-oriented goals for development and implementation of more adequate methods to detect and confirm resistance in the fastest possible method, especially when resistant bacteria are suspected in critical healthcare settings as cancer hospitals and intensive care units. Numerous studies indeed from all over the world detect the resistance genes along with the resistance profile [7–10]. Data about resistance pathogens can never be obsolete due to their big medical importance, and updated reports should always be released as resistance patterns varies in different geographical areas as well as different times.

The conclusions are not completely supported by the data. Although there is a high proportion of resistant strains, it seems difficult to claim that infection control practices are lacking without study data or references about the compliance with these practices at a local or a national level.

Authors’ response:

This was just a genuine remark from our team from our experience of the local infection control practices, however, it was suggested as a definitive conclusion. More data are indeed needed for such deduction, therefore it is removed from the revised manuscript. Lines 387-388 (It also emphasizes the lack of infection control practices) is removed.

References

  1. El-Sokkary, R.H.; Ramadan, R.A.; El-Shabrawy, M.; El-Korashi, L.A.; Elhawary, A.; Embarak, S.; Tash, R.M.E.; Elantouny, N.G. Community Acquired Pneumonia among Adult Patients at an Egyptian University Hospital: Bacterial Etiology, Susceptibility Profile and Evaluation of the Response to Initial Empiric Antibiotic Therapy. Infect Drug Resist 2018, 11, 2141–2150, doi:10.2147/IDR.S182777.
  2. Tadesse, B.T.; Ashley, E.A.; Ongarello, S.; Havumaki, J.; Wijegoonewardena, M.; González, I.J.; Dittrich, S. Antimicrobial Resistance in Africa: A Systematic Review. BMC Infectious Diseases 2017, 17, 616, doi:10.1186/s12879-017-2713-1.
  3. Vijay, S.; Dalela, G. Prevalence of LRTI in Patients Presenting with Productive Cough and Their Antibiotic Resistance Pattern. J Clin Diagn Res 2016, 10, DC09-DC12, doi:10.7860/JCDR/2016/17855.7082.
  4. Abbas, S.; Sabir, A.U.; Khalid, N.; Sabir, S.; Khalid, S.; Haseeb, S.; Numair Khan, M.; Ajmal, W.M.; Azhar, F.; Saeed, M.T. Frequency of Extensively Drug-Resistant Gram-Negative Pathogens in a Tertiary Care Hospital in Pakistan. Cureus 2020, 12, e11914, doi:10.7759/cureus.11914.
  5. Jiang, X.; Zhang, Z.; Li, M.; Zhou, D.; Ruan, F.; Lu, Y. Detection of Extended-Spectrum β-Lactamases in Clinical Isolates of Pseudomonas Aeruginosa. Antimicrob Agents Chemother 2006, 50, 2990–2995, doi:10.1128/AAC.01511-05.
  6. Magiorakos, A.P.; Srinivasan, A.; Carey, R.B.; Carmeli, Y.; Falagas, M.E.; Giske, C.G.; Harbarth, S.; Hindler, J.F.; Kahlmeter, G. Multidrug-Resistant, Extensively Drug-Resistant and Pandrug-Resistant Bacteria: An International Expert Proposal for Interim Standard Definitions for Acquired Resistance. Clinical Microbiology and Infection 2012, 18, 268–281.
  7. Han, R.; Shi, Q.; Wu, S.; Yin, D.; Peng, M.; Dong, D.; Zheng, Y.; Guo, Y.; Zhang, R.; Hu, F.; et al. Dissemination of Carbapenemases (KPC, NDM, OXA-48, IMP, and VIM) Among Carbapenem-Resistant Enterobacteriaceae Isolated From Adult and Children Patients in China. Front. Cell. Infect. Microbiol. 2020, 10, doi:10.3389/fcimb.2020.00314.
  8. Zowawi, H.M.; Sartor, A.L.; Balkhy, H.H.; Walsh, T.R.; Al Johani, S.M.; AlJindan, R.Y.; Alfaresi, M.; Ibrahim, E.; Al-Jardani, A.; Al-Abri, S.; et al. Molecular Characterization of Carbapenemase-Producing Escherichia Coli and Klebsiella Pneumoniae in the Countries of the Gulf Cooperation Council: Dominance of OXA-48 and NDM Producers. Antimicrob Agents Chemother 2014, 58, 3085–3090, doi:10.1128/AAC.02050-13.
  9. Grundmann, H.; Glasner, C.; Albiger, B.; Aanensen, D.M.; Tomlinson, C.T.; Andrasević, A.T.; Cantón, R.; Carmeli, Y.; Friedrich, A.W.; Giske, C.G.; et al. Occurrence of Carbapenemase-Producing Klebsiella Pneumoniae and Escherichia Coli in the European Survey of Carbapenemase-Producing Enterobacteriaceae (EuSCAPE): A Prospective, Multinational Study. The Lancet Infectious Diseases 2017, 17, 153–163, doi:10.1016/S1473-3099(16)30257-2.
  10. Verma, N.; Prahraj, A.K.; Mishra, B.; Behera, B.; Gupta, K. Detection of Carbapenemase-Producing Pseudomonas Aeruginosa by Phenotypic and Genotypic Methods in a Tertiary Care Hospital of East India. J Lab Physicians 2019, 11, 287–291, doi:10.4103/JLP.JLP_136_19.

Reviewer 3 Report

Thank you for the opportunity to review this manuscript.

The manuscript “Correlation between the antibiotic resistant genes and susceptibility to antibiotics among the carbapenem-resistant Gram-negative pathogens” is very up to date and of great interest. The data presented brings light toward resistance encountered in Gram negative bacteria from Egipt. I liked the fact that the authors presented into the Introduction general data about resistance in bacteria and painted a general picture of resistance mechanisms.

I believe the Methods were well presented and the study design well thought. The results are presented from different angles and the discussion compares them with other recent findings. I really enjoyed reading the article.

I couldn’t though stop noticing some spelling mistakes, eg. in row 150 in Table 2 “Sensitivw”.

I do not have other comments to the manuscript, I believe that the work is well organized and I recommend only an English check.

Author Response

Reviewer 3 comments:

The manuscript “Correlation between the antibiotic resistant genes and susceptibility to antibiotics among the carbapenem-resistant Gram-negative pathogens” is very up to date and of great interest. The data presented brings light toward resistance encountered in Gram negative bacteria from Egypt. I liked the fact that the authors presented into the Introduction general data about resistance in bacteria and painted a general picture of resistance mechanisms.

I believe the Methods were well presented and the study design well thought. The results are presented from different angles and the discussion compares them with other recent findings. I really enjoyed reading the article.

Authors’ response:

We could like to express sincere appreciation for the reviewer’s positive feedback.

I couldn’t though stop noticing some spelling mistakes, eg. in row 150 in Table 2 “Sensitivw”.

Authors’ response:

The typographical error was corrected and the whole manuscript was thoroughly revised for any possible typos or grammatic mistakes. However, table 2 was moved to supplementary data as recommended by Reviewer 1. Correction could be found highlighted in the supplementary data file.

I do not have other comments to the manuscript, I believe that the work is well organized and I recommend only an English check.

Authors’ response:

We really appreciate the reviewer’s positive comment and positive conclusion about out study. The manuscript was checked again for more typographical and grammatical mistakes and it was corrected when found.

Round 2

Reviewer 2 Report

To my opinion, the interest of identifying genes of resistance for guiding the clinicians for effective antimicrobial therapy prescription is limited to multiplex PCR panels allowing to provide the presence or absence of certain resistance genes within one or two hours following the respiratory sample in the lab. Thus, it is important to differentiate these rapid methods (e.g. Biofire FilmArray*), which can be useful for de-escalation or for prescribing a wider range antimicrobial therapy within the few hours following the diagnosis of pneumonia, and other molecular methods, which can have an epidemiological interest and can provide information for local or national recommendations. This should be discussed more thoroughly in the discussion section.

Concerning the susceptibility of K. pneumoniae to amoxicillin, P. aeruginosa to co-amoxyclav, etc., I think that the authors should indicate that in certain areas, such isolates had already been identified, with the references.

Author Response

Reviewer 2 comments:

To my opinion, the interest of identifying genes of resistance for guiding the clinicians for effective antimicrobial therapy prescription is limited to multiplex PCR panels allowing to provide the presence or absence of certain resistance genes within one or two hours following the respiratory sample in the lab. Thus, it is important to differentiate these rapid methods (e.g. Biofire FilmArray*), which can be useful for de-escalation or for prescribing a wider range antimicrobial therapy within the few hours following the diagnosis of pneumonia, and other molecular methods, which can have an epidemiological interest and can provide information for local or national recommendations. This should be discussed more thoroughly in the discussion section.

Authors’ response:

This keen comment is genuinely appreciated. We clarified this matter and included in the discussion section (lines 318-338 and 341-342) the following:

Patients suffering from community-acquired pneumonia usually receive empirical antimicrobial therapy while the guidelines reserve the microbiological testing for the severe cases [1]. Standard microbiological identification techniques, followed by antimicrobial susceptibility testing, followed by PCR identification of the resistance genes of concern is a tedious process that requires several days. This delay exposes the patients to the unnecessary adverse effects of the drugs, as well as extending the hospital-stay for the complicated cases which increases the risk that the patients contact a hospital-acquired infection [2]. It is extremely important to implement rapid techniques that allow the identification of the causative pathogens within few hours. This would ensure more effective antimicrobial therapy within few hours following the diagnosis [1]. One of such techniques is the Biofire® FilmArray® Pneumonia Panel which accurately identifies 33 targets in sputum and bronchoalveolar lavage samples in about one hour. It is a multiplex PCR technology that contains probes for 8 respiratory viruses, 18 bacteria as well as 7 clinically relevant resistance genes (mecA/C, blaKPC, blaNDM, blaVIM, blaOXA-48-like, blaIMP and blaCTX-M). This technology identifies the nucleic acids in the samples even if the pathogen is fastidious or the patient received prior antimicrobial therapy which would render the culture results incomprehensive [3]. Other rapid molecular diagnosis techniques include the RespiFinder® SMART 22 FAST, the Unyvero pneumonia cartridge, the ResPlexTM Panels, the Scalable Target Analysis Routine (STAR) technology and the PLEX-ID technology [4–7]. Unfortunately, these techniques are not widespread in the Egyptian hospitals as they are much more expensive.

(Lines 334-335)

Investment to incorporate the rapid identification techniques in the Egyptian hospitals should become a medical priority to allow an improved routine-care.

The addition is found highlighted in the resubmitted manuscript.

Concerning the susceptibility of K. pneumoniae to amoxicillin, P. aeruginosa to co-amoxyclav, etc., I think that the authors should indicate that in certain areas, such isolates had already been identified, with the references.

Authors’ response:

We would like to thank the reviewer for his comment. The following is added to the discussion section in the resubmitted manuscript (lines 266-271).

Although K. pneumoniae is usually resistant to amoxicillin, susceptibility to amoxicillin is still carried out in several countries around the world as not all K. pneumoniae isolates produce penicillinases [8]. This African systematic review included more than 144 studies and 149,000 samples from patients all across Africa. Likewise, several recent studies tested the susceptibility of P. aeruginosa to amoxicillin, co-amoxiclav and cefotaxime as not all P. aeruginosa isolates produce cephalosporinases [8–10].

References:

  1. Gilbert, D.N.; Leggett, J.E.; Wang, L.; Ferdosian, S.; Gelfer, G.D.; Johnston, M.L.; Footer, B.W.; Hendrickson, K.W.; Park, H.S.; White, E.E.; et al. Enhanced Detection of Community-Acquired Pneumonia Pathogens With the BioFire® Pneumonia FilmArray® Panel. Diagn Microbiol Infect Dis 2021, 99, 115246, doi:10.1016/j.diagmicrobio.2020.115246.
  2. Moffa, M.A.; Bremmer, D.N.; Carr, D.; Buchanan, C.; Shively, N.R.; Elrufay, R.; Walsh, T.L. Impact of a Multiplex Polymerase Chain Reaction Assay on the Clinical Management of Adults Undergoing a Lumbar Puncture for Suspected Community-Onset Central Nervous System Infections. Antibiotics 2020, 9, 282, doi:10.3390/antibiotics9060282.
  3. The BioFire® FilmArray® Pneumonia Panel Available online: https://www.biofiredx.com/products/the-filmarray-panels/filmarray-pneumonia/ (accessed on 20 February 2021).
  4. Hattoufi, K.; Tligui, H.; Obtel, M.; El Ftouh, S.; Kharbach, A.; Barkat, A. Molecular Diagnosis of Pneumonia Using Multiplex Real-Time PCR Assay RespiFinder® SMART 22 FAST in a Group of Moroccan Infants Available online: https://www.hindawi.com/journals/av/2020/6212643/ (accessed on 20 February 2021).
  5. Luyt, C.-E.; Hékimian, G.; Bonnet, I.; Bréchot, N.; Schmidt, M.; Robert, J.; Combes, A.; Aubry, A. Usefulness of Point-of-Care Multiplex PCR to Rapidly Identify Pathogens Responsible for Ventilator-Associated Pneumonia and Their Resistance to Antibiotics: An Observational Study. Critical Care 2020, 24, 378, doi:10.1186/s13054-020-03102-2.
  6. Hasan, M.R.; Al Mana, H.; Young, V.; Tang, P.; Thomas, E.; Tan, R.; Tilley, P. A Novel Real-Time PCR Assay Panel for Detection of Common Respiratory Pathogens in a Convenient, Strip-Tube Array Format. J Virol Methods 2019, 265, 42–48, doi:10.1016/j.jviromet.2018.12.013.
  7. Caliendo, A.M. Multiplex PCR and Emerging Technologies for the Detection of Respiratory Pathogens. Clin Infect Dis 2011, 52, S326–S330, doi:10.1093/cid/cir047.
  8. Tadesse, B.T.; Ashley, E.A.; Ongarello, S.; Havumaki, J.; Wijegoonewardena, M.; González, I.J.; Dittrich, S. Antimicrobial Resistance in Africa: A Systematic Review. BMC Infectious Diseases 2017, 17, 616, doi:10.1186/s12879-017-2713-1.
  9. Vijay, S.; Dalela, G. Prevalence of LRTI in Patients Presenting with Productive Cough and Their Antibiotic Resistance Pattern. J Clin Diagn Res 2016, 10, DC09-DC12, doi:10.7860/JCDR/2016/17855.7082.
  10. Abbas, S.; Sabir, A.U.; Khalid, N.; Sabir, S.; Khalid, S.; Haseeb, S.; Numair Khan, M.; Ajmal, W.M.; Azhar, F.; Saeed, M.T. Frequency of Extensively Drug-Resistant Gram-Negative Pathogens in a Tertiary Care Hospital in Pakistan. Cureus 2020, 12, e11914, doi:10.7759/cureus.11914.

All references have been included in the revised manuscript (R2)